# Selectively Targeting Breast Cancer Stem Cells by 8-Quinolinol and Niclosamide

**DOI:** 10.3390/ijms231911760

**Published:** 2022-10-04

**Authors:** Patricia Cámara-Sánchez, Zamira V. Díaz-Riascos, Natalia García-Aranda, Petra Gener, Joaquin Seras-Franzoso, Micaela Giani-Alonso, Miriam Royo, Esther Vázquez, Simó Schwartz, Ibane Abasolo

**Affiliations:** 1Drug Delivery and Targeting Group, Vall d’Hebron Institut de Recerca (VHIR), Universitat Autònoma de Barcelona (UAB), 08035 Barcelona, Spain; 2Networking Research Center on Bioengineering, Biomaterials and Nanomedicine (CIBER-BBN), Instituto de Salud Carlos III (ISCIII), 28029 Madrid, Spain; 3Functional Validation & Preclinical Research (FVPR), Vall d’Hebron Institut de Recerca (VHIR), Universitat Autònoma de Barcelona (UAB), 08035 Barcelona, Spain; 4Institute for Advanced Chemistry (IQAC-CSIC), Jordi Girona 18-26, 08034 Barcelona, Spain; 5Institut de Biotecnologia i de Biomedicina, Universitat Autònoma de Barcelona, 08193 Barcelona, Spain; 6Departament de Genètica i de Microbiologia, Universitat Autònoma de Barcelona, 08193 Barcelona, Spain

**Keywords:** triple negative breast cancer, cancer stem cells, 8-quinolinol, niclosamide, combination therapy

## Abstract

Cancer maintenance, metastatic dissemination and drug resistance are sustained by cancer stem cells (CSCs). Triple negative breast cancer (TNBC) is the breast cancer subtype with the highest number of CSCs and the poorest prognosis. Here, we aimed to identify potential drugs targeting CSCs to be further employed in combination with standard chemotherapy in TNBC treatment. The anti-CSC efficacy of up to 17 small drugs was tested in TNBC cell lines using cell viability assays on differentiated cancer cells and CSCs. Then, the effect of 2 selected drugs (8-quinolinol -8Q- and niclosamide -NCS-) in the cancer stemness features were evaluated using mammosphere growth, cell invasion, migration and anchorage-independent growth assays. Changes in the expression of stemness genes after 8Q or NCS treatment were also evaluated. Moreover, the potential synergism of 8Q and NCS with PTX on CSC proliferation and stemness-related signaling pathways was evaluated using TNBC cell lines, CSC-reporter sublines, and CSC-enriched mammospheres. Finally, the efficacy of NCS in combination with PTX was analyzed in vivo using an orthotopic mouse model of MDA-MB-231 cells. Among all tested drug candidates, 8Q and NCS showed remarkable specific anti-CSC activity in terms of CSC viability, migration, invasion and anchorage independent growth reduction in vitro. Moreover, specific 8Q/PTX and NCS/PTX ratios at which both drugs displayed a synergistic effect in different TNBC cell lines were identified. The sole use of PTX increased the relative presence of CSCs in TNBC cells, whereas the combination of 8Q and NCS counteracted this pro-CSC activity of PTX while significantly reducing cell viability. In vivo, the combination of NCS with PTX reduced tumor growth and limited the dissemination of the disease by reducing circulating tumor cells and the incidence of lung metastasis. The combination of 8Q and NCS with PTX at established ratios inhibits both the proliferation of differentiated cancer cells and the viability of CSCs, paving the way for more efficacious TNBC treatments.

## 1. Introduction

Breast cancer is the most frequent cancer among women, accounting for 24.5% of all new female cancer cases in 2020 worldwide, and overall, it is the leading cause of cancer death in women [1]. The triple negative breast cancer (TNBC) subtype represents approximately 15% to 20% of breast cancers and comprises a heterogeneous group of tumors phenotypically characterized by a lack of expression of estrogen and progesterone receptors and no amplification of HER2 [2,3]. Upon such molecular profile, endocrine or HER2-targeted therapies fail and chemotherapy remains the primary systemic treatment for TNBC [4]. Unfortunately, most patients relapse soon after initial remission, developing metastatic and chemo-resistant diseases with few treatment options. All these factors contribute to TNBC patients being currently the subgroup with the poorest prognosis and worst outcomes among all breast cancer subtypes [2,5].

Among different types of cancers, chemotherapy resistance and metastatic dissemination seem to be sustained by a small subpopulation of cancer cells with stem cell-like properties, termed cancer stem cells (CSCs) [6,7]. These cells have the ability to self-renew, differentiate and progress in very harsh conditions [8]. Interestingly, among the breast cancer subtypes, the highest amount of CSCs was observed in patients with TNBC and has been correlated with its high aggressiveness and higher metastatic potential [9,10]. In breast cancer, CSCs express CD44^+^/CD24^−/low^/linage^−^ stem cell-phenotype and are able to form mammospheres in suspension culture, differentiate into defined progenies, initiate and drive tumor growth in vivo [6,10,11]. Due to the high heterogeneity within CD44^+^/CD24^−/low^/lineage^−^ population, additional stem cell-like markers have been defined that help to isolate or enrich CSCs by surface markers (i.e., CD133, CD49f, CD61 and EpCAM), intracellular enzymatic activities (i.e., Aldefluor) or the ability to pump out dies (i.e., side population by Hoechst 33342) [6,7,10]. Particularly relevant is the abnormally enhanced enzymatic activity of aldehyde dehydrogenase 1 (ALDH1), specifically ALDH1A1, in CSCs [7,8]. The use of a minimal ALDH1A1 promoter linked to a fluorescent reporter has allowed the tracking of the stemness phenotype in cancer cells, as well as their enrichment and treatment assessment by flow cytometry [12,13]. In this model, cells expressing the fluorescent reporter also expressed different CSC markers, such ALOX5, Oct4, Nanog, Notch4 or the ATP-binding cassette (ABC) transporter G family ABCG2, further validating the usefulness of this cell model [10,12,13,14].

Overall, the identification of surface markers enriched in CSCs has allowed the design of drug delivery systems specifically targeting the CSC population [15,16,17]. However, for completely eradicating cancer, it is not only important to deliver drugs into CSCs but to deliver the appropriate drug. CSCs are relatively quiescent, enabling them to withstand otherwise lethal doses of chemotherapy and survive to promote tumor recurrence [15]. Specific signaling routes, known to be very active in stem cells, are also activated in CSCs. Particularly, Notch, Wnt/β-catenin, PI3K/Akt/mTOR, NF-κB, HIF and STAT3 pathways have been found constitutively activated in breast cancer, and in particular, in TNBC, and their deregulation contributes to the maintenance of CSCs [8,10,18]. Thus, different small drugs targeting these signaling routes have been tested as potential anti-cancer treatments with relative success [8]. For instance, Notch inhibition results in a reduction of CD44^+^/CD24^−^/low CSC population and tumor growth by sensitizing CSC to anticancer therapies [19,20]. Similarly, the use of a specific Wnt pathway inhibitor reduces both CD44^+^/CD24^−^/low and ALDH+ breast CSCs and inhibits the self-renewal and metastasis of this cell population [21,22].

In this work, we hypothesized that the use of specific anti-CSC agents in combination with conventional chemotherapeutic drugs may increase the efficacy of treatment in advanced TNBC by targeting both CSCs and bulk tumor cells. With this aim, we first screened the efficacy of up to 17 different anti-CSC drugs in TNBC cell lines and CSC models, and then studied the potential synergism of selected anti-CSC drugs with the anti-neoplastic reference drug paclitaxel (PTX) using in vitro and in vivo models of TNBC.

## 2. Results

### 2.1. Selection of Candidates with High Anti-Proliferative Activity in Human Breast Cancer Cell Lines

A total number of 17 approved small drugs were selected for testing their activity in breast cancer cells (see Table 1). All selected compounds were previously described as interfering with essential signaling pathways related to CSC properties. Chemical structures and references to previous publications are shown in Appendix A. We first evaluated the in vitro cytotoxicity of test compounds by MTT in the basal-like MDA-MB-231 and luminal A MCF-7 cell lines and compared it to PTX, an anti-neoplastic drug used as a first-line treatment in TNBC. Out of the total of screened compounds, six displayed high cytotoxic potency with IC_50_ values below 10 µM in MDA-MB-231 and MCF-7 cell lines, namely YM, PNB, VS, NCS, NTC and SAL (Table 1), close to the values obtained for the reference drug PTX. As expected, MCF-7 cells were more sensitive (lower IC_50_ values) to the tested compounds than MDA-MB-231 cells, known to be highly chemo-resistant and aggressive [23] (Table 1 and Appendix A). Furthermore, we thought it was interesting to evaluate whether the anti-proliferative activity of drugs obtained was similar in other TNBC cell lines. To this end, of all drugs with a good anti-proliferative profile, four (YM, PNB, NCS and FLU) were selected due to their combined low IC_50_ values in MCF-7 and MDA-MB-231 cell lines. These four compounds, along with 8Q, were further tested in additional MTT assays in an extended battery of breast cancer cell lines. As expected, when IC_50_ values from the different TNBC cell lines were compared, the drug cytotoxic efficiency differed substantially depending on the tested cell line (Table 1).

### 2.2. Identification of Candidates with Selective Anti-CSC Activity

Selected drugs were then subjected to proliferation assays in CSC and non-CSC subpopulations sorted from the MDA-MB-231-ALDH1A1:tdTomato reporter cell line [13]. Sorted cells were characterized by the expression of stemness markers (Appendix A) and further subjected to conventional proliferation. No significant differences in treatment sensitivity were observed between CSCs and non-CSCs when treated with the reference drug PTX, thereby confirming that the chemotherapeutic agent did not show a selective cytotoxic effect against CSCs (Figure 1A). Moreover, 8Q and NCS were the only drugs that exhibited significantly higher sensitivity in CSCs compared to non-CSCs (Figure 1B). Indeed, IC_50_ values for 8Q and NCS drugs were 1.33 and 3.66-fold lower (*p* < 0.05) in the CSC in comparison with the non-CSC subpopulation, respectively (Figure 1A,B). The results for SAL also moved in the same direction, but differences among IC_50_ values of CSCs and non-CSCs were not statistically significant. Conversely, NTC, VS, DSF, FLU and DFT drugs showed no differences in cell viability between CSCs and non-CSCs and were discarded.

The effect of 8Q, NCS and SAL on CSCs was further studied in low attachment conditions, where mammosphere formation (MSF) is promoted. As shown in Figure 2A, the inhibition of mammosphere-forming efficiency increased in a dose-dependent manner in all three anti-CSC drugs, being higher in the case of 8Q and NCS (Figure 2A), which prevented MSF in 27.4% ± 6.7% and 13.8% ± 7.5%, respectively, when the highest concentration was tested (Figure 2B). Accordingly, these were the compounds that showed the lowest IC_50_ values, 2.71 µM ± 0.14 for 8Q and 1.65 µM ± 0.79 for NCS. Significantly, the IC_50_ value of 8Q in low attachment conditions was lower than the IC_50_ value in conventional cultures (25.55 ± 0.151 µM, data in Table 1). On the other hand, the effect on MSF was remarkably lower for SAL (IC_50_ of 4.95 ± 0.83 µM) in comparison with 8Q and NCS, but significantly higher (*p* < 0.0001) than that produced by the anti-neoplastic reference drug PTX (Figure 2A,B). Indeed, the higher dose tested for PTX (2 µM, twice the IC_50_ value in conventional cultures) did not reach 50% mammosphere growth inhibition. These results pointed out that PTX was not as effective as 8Q, NCS or SAL in selectively eradicating CSC populations. Moreover, given that 8Q and NCS showed the highest efficacy in inhibiting MSF, these two drugs were chosen to continue with functional assays.

### 2.3. Q and NCS Affect Specific Stem Cell-Like Features of Breast CSC Subpopulation

We next examined whether 8Q and NCS impacted other distinctive features of CSCs, including migration, anchorage-independent growth (neoplastic transformation) and cell invasion ability, using the MDA-MB-231 cell line (Figure 3). Regarding migration, significantly lower wound closure was observed in cells treated with anti-CSC drugs compared to the controls (Figure 3A,B). At 24 h, the open wound detected for the control samples was almost minimal (4.45% ± 1.12%), while in cells treated with 8Q and NCS, it was 33.9% ± 1.8% and 24.9% ± 1.7%, respectively (Figure 3A). Of note, treatment with PTX also showed a high inhibitory effect on cell migration (Figure 3A,B). Interestingly, anchorage-independent growth of MDA-MB-231 cells in soft agar was significantly inhibited by treatments with 8Q and NCS (Figure 3C). At the highest tested concentrations of 8Q (50 µM) and NCS (15 µM), the capacity of malignant cells to form colonies was significantly reduced to 5.40% ± 1.23% and 11.48% ± 1.11% of the growth observed in the non-treated cells, respectively. Conversely, PTX treatment was not able to reduce the anchorage-independent growth below 50%, thereby demonstrating that 8Q and NCS had superior anti-CSC activity (Figure 3C).

Furthermore, we explored the effects of 8Q and NCS on the invasiveness of stem cell-like cells. The results showed that 8Q significantly reduced the ability of cells to cross the Matrigel-covered 8-µm filter, leading to almost complete inhibition of cell invasion ability (Figure 3D). In the case of NCS, no inhibitory effect was observed in the cell invasion assays. PTX also showed some efficacy in reducing cell invasion, but it was significantly lower than that obtained with 8Q (Figure 3D). Altogether, the results demonstrated that 8Q and NCS had strong anti-CSC activity in MDA-MB-231 cells by inhibiting crucial stemness features of CSCs.

### 2.4. Q and NCS Display a Synergic Anti-Proliferation Effect When Combined with PTX at Specific Ratios in Different TNBC Cell Lines

Since anti-CSC drugs are not meant to be used solely in cancer treatment, the combination of 8Q and NCS in combination with PTX was evaluated on MDA-MB-231, HCC-1806 and MDA-MB-468 cell lines by MTT assays. Depending on the drug doses, a combination can yield a synergistic, additive or antagonistic effect; thus, the combination index (CI) for each drug interaction was estimated for different drug ratios (Figure 4). To better determine the potential synergism of the drugs, assays were conducted by fixing each drug at its IC_50_ value and varying the other one. For the PTX-8Q combination, an enhanced cytotoxic effect was observed at multiple PTX:8Q ratios, of which 1:12.5 was the one that resulted in the highest synergistic effect with the lowest CI value (CI = 0.06) (Figure 4A,B). For the PTX-NCS combination, the greatest drug synergy was observed at a 1:2 ratio of PTX:NCS with a CI value close to 0, thus indicating a strong positive synergism between both drugs (Figure 4C,D). Accordingly, these ratios were used in later studies on MDA-MB-231 cells.

Combination assays of 8Q and NCS with PTX in HCC-1806 and MDA-MB-468 cell lines also showed improved cytotoxicity when combined at specific ratios (see Appendix A). In HCC-1806 cells, the greatest synergy was obtained at 1:5 (with a CI close to 0) and 1:0.4 (CI = 0.17) ratios for the PTX-8Q and PTX-NCS treatments, respectively. In MDA-MB-468 cells, it was obtained at 1:1250 (CI = 0.02) and 1:1000 (CI = 0.12) drug ratios. The differences in the established optimal ratios between these cell lines are likely a reflection of differential molecular and intracellular signaling changes, as well as differential drug sensitivity to treatments tested, demonstrating the cell-type-specific response to combined therapy.

### 2.5. Combination of 8Q or NCS with PTX Increases the Anti-CSC Efficacy of the Drugs

To evaluate whether combination therapy of PTX-8Q and PTX-NCS could offer therapeutic advantages, drug ratios with the best CI values were evaluated in the fluorescent tdTomato CSC models. Changes in the CSC subpopulation were monitored by flow cytometry and stem cell gene expression analysis (Figure 5 for the MDA-MB-231 cell line and Appendix A for MDA-MB-468 and HCC-1806). As expected, following PTX exposure, the relative abundance of CSCs (tdTomato+ cells) significantly increased (*p* < 0.0001) in a dose-dependent manner in MDA-MB-231 cells (Figure 5A). Conversely, upon incubation with 8Q and NCS anti-CSC drugs, the relative amount of MDA-MB-231-tdTomato+ cells remarkably decreased (*p* < 0.0001 for both drugs) when compared to PTX treatment. Of note, low doses of 8Q achieved a significant reduction in the MDA-MB-231 CSC subpopulation, while at higher doses, the effect on CSCs was only preventive. In the case of NCS treatment, such a decrease in CSC-tdTomato+ was dose dependent. Interestingly, combined treatments of 8Q and NCS with PTX significantly abrogated the relative increase of MDA-MB-231-tdTomato+ cells induced by PTX, as well as significantly enhanced the anti-CSC effect of 8Q and NCS individual treatments (*p* = 0.0201 and *p* = 0.003 for lower doses of PTX-8Q and PTX-NCS, respectively, while for higher doses was *p* < 0.0001 for both combinations; compared to individual anti-CSC therapy).

These results were further confirmed in the MDA-MB-468 and HCC-1806 TNBC cell lines (Appendix A). Upon incubation of TNBC cell lines with 8Q and NCS anti-CSC drugs, the relative presence of tdTomato+ cells was significantly reduced (*p* < 0.0001; for both drugs in both cell lines) when compared to PTX treatment. In the case of NCS treatment, such a decrease was more evident and dose-dependent in HCC-1806 and MDA-MB-468 cell lines than in MDA-MB-231 (Appendix A). Furthermore, the anti-CSC effects of 8Q and NCS were significantly enhanced when drugs were administered in combination with PTX rather than as an individual therapy. Interestingly, the extent of the reduction was similar to that observed in MDA-MB-231 cells, moving from values at 1.51+/−0.04 and 1.31+/−0.03 for PTX in MDA-MB-468 and HCC-1806, respectively, to a mean value of 0.63+/−0.02 and 0.66+/−0.18 for PTX combined with 8Q and NCS, respectively (compared to 0.67+/−0.01 and 0.75+/−1.2 in MDA-MB-231). However, when looking at the effect of specific CSC genes (Appendix A), the combination of 8Q for NCS with PTX seems to be more effective combined treatments in MDA-MB-468 and HCC-1806 cells than in MDA-MB-231.

These results were further confirmed by gene expression analysis in all three TNBC cell lines. In MDA-MB-231 cells, we found a remarkable relative increase in the expression of all stem cell genes analyzed, including ALDH1A1, ALOX5, CMKLR1, ABCg2, Notch4, Nanog and OCT4 after PTX individual treatment (Figure 5B). Meanwhile, the 8Q and NCS treatments, either alone or in combination with PTX, substantially downregulated the expression of stem cell genes. In particular, 8Q single treatment led to a strong decrease in ALDH1A1, ALOX5 and ABCG2 mRNA levels, while for the other genes, the effect obtained was slighter. In the case of NCS treatment, such a decrease was greater in ALDH1A1, ALOX5 and Nanog mRNA, and indeed, was stronger than when treated with 8Q. The combination of both drugs with PTX remarkably downregulated the expression of almost all stem cell-like markers analyzed, such as ALDH1A1, ALOX5, Notch4 and Nanog, thus indicating preferential and efficient effects against CSCs when 8Q and NCS were used in combination with PTX (Figure 5B). These results were further confirmed in the HCC-1806 and MDA-MB-468 cell lines (Appendix A). Importantly, in both cell lines, a prominent increase in stem cell gene expression was also obtained after PTX treatment. Such an increase was stronger in Notch4 and OCT4 mRNA than in the MDA-MB-231 cell line. Moreover, 8Q and NCS, either alone or in combination with PTX, remarkably downregulated stem cell gene expression in both cell lines, overcoming and, in some cases, only matching the effect of individual treatments in almost all genes analyzed.

The effect of the combination of 8Q and NCS with PTX on the CSC subpopulation was further confirmed by analyzing their effect on mammosphere viability (MSV) and the ability to form new ones (mammosphere-forming efficiency, MSF) in all three TNBC cell lines (Figure 6 and Appendix A). Consistent with previous results, the ability of MDA-MB-231 stem-like cells to grow as mammospheres was significantly reduced when treated with 8Q and NCS (*p* < 0.0001 in both cases). This reduction was significantly greater when drugs were used in combination (*p* = 0.0014 and *p* = 0.0101 for combinations of PTX with 8Q and NCS, respectively) than in single treatments (Figure 6A). The anti-neoplastic drug PTX showed a limited impact on mammosphere growth, since the PTX dose did not succeed in reducing to 50% the MSF (75.9% ± 4.7%). Similar results were obtained when MSV was analyzed in MDA-MB-231 (Figure 6B), where the effect of PTX was limited (85.5% ± 4.7%), but the use of 8Q and NCS induced a relevant loss of mammosphere viability, especially when combined with PTX (*p* = 0.0005 and *p* = 0.0186 for combinations with 8Q and NCS, respectively). The described results were confirmed in the HCC-1806 and MDA-MB-468 TNBC cell lines (Appendix A). In these cells, PTX showed no impact on MFA or MSV (65.9% ± 4.3% and 83.3% ± 2.6% in MSF for HCC-1806 and MDA-MB-468 cells, respectively, while in MSV, it was 92.4% ± 3.9% and 86.7% ± 5.1%, respectively). On the contrary, the ability of stem-like tumor cells to grow as mammospheres from both TNBC cell lines was significantly reduced when treated with 8Q and NCS (*p* < 0.0001 for both drugs). Such reduction was significantly greater when drugs were used in combination (*p* < 0.0001 and *p* = 0.0031 for combinations of PTX with 8Q and NCS in HCC-1806 cells, respectively; *p* < 0.0001 and *p* = 0.0043 for MDA-MB-468 cells, respectively) than as single treatments (Appendix A). Similar findings were observed when MSV was analyzed (Appendix A). The use of 8Q and NCS induced a relevant loss of mammosphere viability, especially when combined with PTX (*p* = 0.0069 and *p* = 0.0014 for combinations with 8Q and NCS in HCC-1806, respectively; *p* = 0.004 and *p* = 0.0046 for MDA-MB-468 cells, respectively). Altogether, NCS showed a superior effect in comparison to 8Q treatment, either alone or in combination with PTX, regarding mammosphere formation and growth.

### 2.6. The Combination of 8Q or NCS with PTX Inhibits NF-κB and Wnt/β-Catenin Signaling Pathways

8Q and NCS have been described as inhibitors of nuclear factor-κB (NF-κB) and Wnt/β-catenin signaling pathways, respectively. Both pathways are known to be crucial in maintaining the stem phenotype [24,25]. In order to elucidate whether the synergism observed on cell viability between 8Q and NCS and the PTX was sustained by alterations in these two pathways, MDA-MB-231 cells were treated with 8Q, NCS and PTX for 24 h, and signaling protein levels were assessed by Western blot. The results pointed out that a significant reduction of phospho-NF-κB (p-NF-κB) subunit p65 was observed for 8Q concentrations as low as 12.5 μM, while the total expression of NF-κB subunit p65 was not changed, thus confirming the role of the 8Q drug as a selective inhibitor of the NF-κB signaling pathway (Appendix A). As expected, PTX treatment showed no efficacy in inhibiting the NF-κB signaling pathway. Moreover, a marked reduction of the p-NF-kB p65 subunit was also observed in cells treated with the combination compared to non-treated (control) or PTX-treated cells, and the inhibition of NF-kB p65 phosphorylation with the PTX+8Q combination was as efficient as with the 8Q alone (Figure 7).

As for the NCS, MDA-MB-231 cells treated with the drug showed a significant dose-dependent reduction of β-catenin protein level as well as an increase of phosphorylated GSK3-β (p-GSK3-β) protein (while total expression of GSK3-β was not changed) (Figure 8 and Appendix A). Conversely, PTX treatment showed no efficacy in inhibiting the Wnt signaling pathway (Figure 8). Importantly, the Western blot analyses confirmed a significant increase of p-GSK3-β and reduction of β-Catenin proteins in TNBC cells when treated with combined therapy (Figure 8).

### 2.7. The Combination of PTX with NCS Enhances Abrogates the In Vivo Growth of MDA-MB-231 Tumors

As shown in previous in vitro studies, lower doses of NCS are needed to obtain more significant anti-CSC activity than 8Q. Therefore, we selected NCS to further test the effect of its combination with PTX in vivo. In detail, NOD/SCID mice with orthotopic MDA-MB-231 tumors were administered with the vehicle, PTX or the combination of PTX with NCS. The treatment schedule followed was established in accordance with previous tests performed in order to reduce side effects without compromising therapeutic efficacy. Tumor growth during the treatment period and the number of circulating tumor cells and lung metastasis at the end-point were evaluated. As for tumor growth, the combination of PTX with NCS had the strongest effect on reducing MDA-MB-231 tumor growth, resulting in smaller tumor volumes (Figure 9A) and weights (Figure 9B). These significant differences were already visible on day 9 and increased throughout the experiment (Figure 9A). More specifically, analysis of excised tumors showed that combined treatment with PTX-NCS reduced both tumor volumes and weights 3.1-fold after co-treatment. In addition, the mice seemed to tolerate treatments without overt severe signs of toxicity or excessive loss of body weight (Appendix A).

### 2.8. The Combination of PTX with NCS Reduces the Presence of Circulating Tumor Cells (CTCs) and the Presence of Lung Metastasis In Vivo

The CTCs in the mice bloodstream at the treatment endpoint (day 24) were analyzed in order to evaluate the effect of the combination in restraining metastatic spread. Although the effect of PTX in primary tumors was clear, PTX-only treatment showed no significant effect on reducing the CTC population when compared to the vehicle (Figure 10A). Meanwhile, flow cytometry analysis revealed that animals treated with the PTX-NCS combination effectively eliminated the CTCs in comparison to single PTX treatment or vehicle (*p* = 0.0070 and *p* < 0.0124, respectively), reducing 2.27- and 2.10-fold, respectively, the CTC content (Figure 10A). Additionally, the presence of lung metastasis was evaluated by ex vivo BLI of the lungs. An example of the BLI images acquired from the excised lungs of all three study groups at endpoint is shown in Figure 10B All animals treated with vehicle (6/6) or PTX-only (9/9) showed positive BLI signals in the lungs (100% incidence), while this incidence dropped down to 67% (6/9 animals) in the case of animals treated with the combination. Moreover, when lung BLI intensity was analyzed, the combination treatment showed a significant decrease in the BLI signal when compared to the single PTX treatment (*p* = 0.0492) or the vehicle (*p* = 0.0087), reducing 8.56- and 42.83-fold lung BLI intensity, respectively (Figure 10B).

## 3. Discussion

The identification of the CSC population has undoubtedly provided a crucial breakthrough in the understanding of cancer biology. These cells have self-renewal capacity and differentiation potential and contribute to multiple tumor malignancies, such as recurrence, metastasis, heterogeneity, multidrug resistance, and radiation resistance [26,27]. Accordingly, CSCs are considered a promising therapeutic target for developing efficient therapies for breast cancer treatment. Such therapies are particularly relevant for TNBC patients due to the lack of specific treatments and the high content of CSC populations in TNBC tumors [9]. Currently, conventional chemotherapy is a standard treatment for TNBC, but it spares CSC populations, which cause tumor recurrence and progression [28]. In recent decades, multiple CSC-targeting strategies have shown therapeutic potential for TNBC in multiple preclinical studies [29], and although some of these strategies are currently being evaluated in clinical trials, such as the AKT inhibitor MK-2206 [30] and the chemokine receptor I/II inhibitor reparixin [31], no therapy specifically targeted against CSCs has already been approved for TNBC treatment. Therefore, the identification of novel therapeutic anti-CSC agents is urgently needed to develop effective therapeutics for this cancer subtype. In this work, we focus on the validation of potential anti-CSC drugs that could inhibit CSC proliferation and downregulate signaling pathways, keeping the stemness phenotype in cancer cells. For this purpose, a battery of 17 compounds was initially identified, all clinically approved drugs with previously described anti-CSC activity (Appendix A). Of note, repurposing approved drugs is a cost-effective strategy that can bypass the time-consuming stages of drug development and thus facilitate rapid clinical translation [32].

From the initial drug set, 8Q and NCS compounds emerged, showing remarkably anti-proliferative activity on the CSC subpopulation, either in attachment or in low attachment culture conditions. It is important to note that we did not select the compounds according to their potency but to their selectivity. SAL, for instance, had lower IC_50_ values against CSC than 8Q, but the effect of the latter was more specific to CSC. In addition, we further demonstrated that 8Q and NCS efficiently inhibited different stemness properties, i.e., migration, invasion and neoplastic transformation. These results are consistent with the works published by Wang et al. [25] and Zhou et al. [24], in which 8Q and NCS were identified as having preferential activity against breast cancer sphere cells, showing their effect through the downregulation of stem pathways and inducing CSC apoptosis. In agreement with previously published studies [33,34], the reference drug PTX was not capable of selectively targeting the CSC subpopulation, either in attachment or in low attachment conditions.

Traditional chemotherapies have failed to remove CSCs and therefore, they have proved ineffective in preventing tumor recurrence. In this scenario, anti-CSC drug administration in combination with traditional chemotherapy may ameliorate clinical outcomes and long-term patient survival. Furthermore, combination therapy is a novel strategy widely used in clinical studies for cancer treatment. It pursues the achievement of a high synergistic therapeutic efficacy at lower drug doses, namely, increasing the treatment efficacy while reducing non desired side-effects [35]. In this scenario, we hypothesized that the use of specific anti-CSC drugs in combination with current anti-cancer reference agents may become a real therapeutic alternative to improve the treatment of TNBC. Following this hypothesis, the synergistic effect of both anti-CSC drugs 8Q and NCS with the anti-cancer drug PTX was explored through combination assays in vitro in different TNBC cell lines. Our data provided strong evidence that both anti-CSC drugs displayed synergistic anti-proliferation activity with PTX when combined at specific ratios. It is worth noting that the synergistic ratios were drug and cell line dependent, probably due to the molecular and phenotypic heterogeneity among the different cell lines. In this regard, such differences evidence the complexity and heterogeneity of cancer, especially in the case of TNBC. There are many ongoing efforts to understand this inherent variability and resistant nature of cancer, as well as to characterize the molecular differences between tumors, with the aim of developing specialized treatments for each specific subtype of cancer, mainly based on the measurement and manipulation of key patient genetic and omics data (transcriptomics, metabolomics, proteomics, etc.) [36]. The understanding and application of these data as tools in clinical trial design and treatment selection have steered the field of cancer treatment toward the concept of precision and personalized medicine, in which therapy selection is tailored to each individual [36,37]. An example of such a personalized approach is the well-known drug trastuzumab, which was approved years ago for the treatment of HER2 receptor positive breast cancer [38], but also the promising use of the poly-ADP ribose polymerase inhibitor olaparib in the treatment of BRCA-mutant ovarian cancer [39]. Accordingly, clinical implementation of drug combination approaches should be considered in a personalized context for the individual patient, ensuring maximum performance and getting the most out of combined therapy in each case. Several studies have already demonstrated the safety, feasibility and importance of designing precision oncology trials that emphasize personalized, individually tailored combination therapies rather than scripted monotherapies. The targeting of a larger fraction of identified molecular alterations has been correlated with significantly improved disease control rates, as well as with longer progression-free and overall survival rates, compared to targeting just one driver mutation, according to a study published by Jason K. Sicklick et al. [40]. In this work, the researchers demonstrated that the use of multi-drug therapies helped improve outcomes among patients with therapy-resistant cancers, indicating that combination drug treatments could improve precision medicine for cancer care. It is clear that integrating a PPM perspective into cancer research and tumor treatment could result in major improvements in fighting cancer, especially due to its complexity and interpatient variability.

Our findings in combination studies showed how two drugs could move from acting synergistically to being antagonistic depending solely on their relative doses. In this regard, optimization of the customized drug ratio is vital prior to further treatment implementation. Interestingly, in our hands, the combination of 8Q with NCS abrogated the relative increase in CSCs induced by PTX. In TNBC cells, 8Q and NCS inhibited NF-κB and Wnt/β-catenin signaling pathways. Both pathways have been reported to be overexpressed in CSC subpopulations, promoting proliferation and resistance to therapy and preserving the undifferentiated state of stem cells [8,10]. An enhanced inhibition of both pathways was obtained when cells were treated with established ratios of PTX with 8Q and NCS, indicating that combination therapy resulted in the suppression of bulk tumor cell proliferation but also enhanced the sensitivity of chemo-resistant cells. Based on these results, we moved forward in testing the efficacy of PTX-NCS treatment in vivo in an orthotopic TNBC mice model. A number of studies have evaluated the potential of NCS as an anti-cancer therapy in various cancer types in vivo, including TNBC [41,42,43,44] and stated the ability of NCS to overcome cancer chemoresistance when combined with PTX [45,46,47] and other anti-cancer agents [48,49]. However, the synergistic effect of combined treatment in CSC subpopulations has not yet been clearly elucidated. In the present study, we showed that the combined treatment of PTX and NCS effectively arrested MDA-MB-231 tumor growth in the TNBC cancer xenograft model and significantly reduced the metastatic ability of aggressive tumor cells, indicating that combined therapy not only improved the therapeutic efficacy of treatments in differentiated tumor cells but also in the CSC subpopulation. Such synergism in MDA-MB-231 cells is mediated via Wnt/β-catenin pathway, and probably executed through the inhibition of Bcl-2 protein [50,51]. Importantly, combination therapy successfully decreased the number of CTC in blood and, more importantly, prevented the generation of lung metastasis, results not observed following treatment with the PTX drug alone.

Based on our findings, treatment with PTX combined with NCS may offer an effective therapeutic approach to improving the prognosis of TNBC by simultaneously targeting both bulk differentiated cancer cells and the minor population of CSCs. Personalized treatment with combination therapies may yield novel chemotherapy strategies in the future.

## 4. Methods and Materials

### 4.1. Pharmacological Agents

The compounds used in this thesis were selected based on their reported activity against CSC, their presence in clinical trials and their commercial availability. First, a literature screening was carried out in PubMed using the terms ‘breast cancer stem cells,’ ‘targeting’ and ‘triple negative breast cancer.’ The objective was to find compounds with specific activity against CSC to further demonstrate their activity in our CSC models. Seventeen compounds were finally selected. Molecular structures, as well as references to previous works, have been detailed in the Appendix A. 8-quinolinol (8Q), acetaminophen (ACE), citral (CIT), disulfiram (DSF), everolimus (EVE), glabridin (GLA), metformin hydrochloride (MET), niclosamide (NCS), nitidine chloride (NTC), paclitaxel (PTX) and salinomycin (SAL) were purchased from Sigma-Aldrich (St. Louis, MO, USA). Defactinib (DFT), panobinostat (PNB) and VS-5584 (VS) were obtained from Selleck Chemicals (Houston, TX, USA). 6-shogaol (6-SHO), flubendazole (FLU) and YM-155 (YM) were purchased from MedChemExpress (Monmouth Junction, NJ, USA). Isoliquiritigenin (ISO) was obtained from LC laboratories (Woburn, MA, USA). Stock solutions were prepared using DMSO or water as solvent and stored as recommended by the manufacturer. All drugs were diluted to the final concentration in a fresh culture medium on the day of the experiments.

### 4.2. Cell Lines and Culture Conditions

Human breast cancer cell lines MDA-MB-231, MCF-7, BT-549, BT-20, MDA-MB-468 and HCC-1806 were obtained from American Type Culture Collection (ATTC, LGC Standards, London, UK). MDA-MB-231 and MCF-7 cell lines were cultured in DMEM/F-12 medium, BT-549, BT-20 and HCC-1806 cells in RPMI 1640 medium and MDA-MB-468 in DMEM high glucose medium (all from Gibco, Thermo Fisher Scientific, Madrid, Spain). All media were supplemented with 10% fetal bovine serum (FBS; Gibco, Thermo Fisher Scientific, Madrid, Spain), 1% antibiotic-antimycotic mixture, 2 mM L-glutamine, 1× non-essential amino acids and 1 mM of sodium pyruvate (all from Gibco, Thermo Fisher Scientific, Madrid, Spain). MDA-MB-231-, HCC1806- and MDA-MB-468-ALDH1A1:tdTomato CSC models, in which CSC can be identified by the expression of tdTomato fluorochrome [13], were cultured in complete medium supplemented with the selection antibiotic blasticidin at 1 μg/mL (Gibco, Thermo Fisher Scientific, Madrid, Spain). Of note, the stemness nature of tdTomato-expressing cells from all three-CSC models has already been fully confirmed in vitro by increased expression of stemness markers, mammosphere formation and in vivo tumorigenic capacity [13]. All cell lines were incubated in a humidified incubator at 37 °C containing 5% CO_2_.

### 4.3. Cytotoxicity Assays (MTT)

Cell viability after 72 h incubation with different drug concentrations was evaluated by MTT ((3-(4,5-dimethylthiazol-2-yl)-2,5-diphenyltetrazolium bromide) tetrazolium reduction assay, following previously described procedures [52]. Cell viability was referred to that in non-treated controls and thereafter the half maximal inhibitory concentration (IC_50_) index was calculated using GraphPad Prism 6 software (GraphPad Software Inc., San Diego, CA, USA). For drug synergic combination analysis, the combination index (CI) value was calculated using CompuSyn software (ComboSyn Inc., Paramus, NJ, USA), based on the multiple drug-effect equation of Chou and Talalay [53], where CI values < 1, =1, or >1 would mean synergistic, additive or antagonistic activity, respectively.

### 4.4. Mammosphere Formation Assay

The ability of selected drugs to affect mammosphere-forming efficiency (MSF) and mammosphere viability (MSV) was studied in ultra-low attachment 96-well plates (Corning^®^, One Riverfront Plaza Corning, Corning, NY, USA). Cells were plated at low density (1000 viable cells/well) in serum-free RPMI 1640 media supplemented with 2% antibiotic-antimycotic mixture and additional factors enhancing CSC maintenance [12,20]. In detail, glucose (60 mg/mL), L-glutamine (10 µL/mL), heparin (4 µg/mL), BSA (2 mg/mL), EGF (0.02 µg/mL), FGFb (0.01 µg/mL), putrescin (10 µg/mL), apo-transferrin (0.1 mg/mL), insulin (25 µg/mL), 30 µM selenium and 20 µM progesterone (all from Sigma-Aldrich, St. Louis, MO, USA). Twenty-four hours after seeding, the cells were treated with drugs for 7 days. The formed spheres were observed by the optical microscope (Olympus IMT-2 model) and quantified by presto blue (Invitrogen, Thermo Fisher Scientific, Madrid, Spain) to further calculate IC_50_ as previously described [12,25].

### 4.5. Wound Healing Assay

The effect of drugs on the migratory potential of MDA-MB-231 cells was studied by recording with a microscope (FSX100 microscope, Olympus Life Science) the closure of a scratch made in confluent monolayers [54]. Wound area was measured at 0, 8 and 24 h by image analysis with ImageJ software [55]. To exclude the effect of cell proliferation in wound healing, treatment doses (1 µM PTX, 25 µM 8Q and 0.5 µM NCS) corresponded to the maximum dose, keeping cell viability above 80% after 24 h incubation.

### 4.6. Colony Formation Assay (Anchorage-Independent Growth) in Soft Agar

Anchorage-independent growth was assessed using CytoSelect™ Cell Transformation Assay Kit (Cell Biolabs Inc., San Diego, CA, USA). A semisolid agar medium was prepared according to the manufacturer’s instructions. Selected drugs were added, and the plates were incubated for 6–8 days at 37 °C in a 5% CO_2_ incubator. Following treatment, colonies formed were observed under an optical microscope and viable transformed cells were quantified by measuring cell viability by MTT at 570 nm.

### 4.7. Matrigel Cell Invasion Assay

Invasion experiments were performed using a conventional 24-well plate with cell culture inserts containing 8 µM pore size filters (Falcon^TM^, Fisher Scientific, Madrid, Spain) and coating the filters with the basement membrane Matrigel (1 mg/mL, BD Biosciences, Franklin Lakes, NJ, USA). After Matrigel solidification, MDA-MB-231 cells (10,000 cells/300 µL, with previous 24 h of FBS starvation) were added into the upper chamber of each insert together with the selected treatments. The chemo-attractant (complete medium with 10% FBS) was placed in the lower chamber of each well. After 24 h incubation at 37 °C in a 5% CO_2_ incubator, the upper surface of the filter was wiped with a cotton-tipped applicator to remove non-invading cells. Cells that had invaded through the filter pores and attached to the under surface of the filter were fixed with 100% methanol and stained with 0.4% crystal violet (Sigma-Aldrich, St. Louis, MO, USA) solution for 15 min. The membranes were mounted on glass slides, and cells from 10 random microscopic fields (20× magnifications) were counted using ImageJ software [56].

### 4.8. Fluorescence-Activated Cell-Sorting Enrichment

A cell sorting assay was performed to select and expand CSCs that expressed tdTomato fluorophore under the ALDH1A1 promoter [12,13]. Briefly, TNBC cells stably transfected with pLenti6-ALDH1A1-tdTomato plasmid (MDA-MB-231-, HCC1806- and MDA-MB-468-ALDH1A1:tdTomato cells) were harvested and suspended in PBS with 10% FBS and 10 µg/mL DAPI and filtered using 30 μm sterile filters (CellTrics^®^, Sysmex Europe GmbH, Norderstedt, Germany) before subjecting to cell sorting with the BD FACSAria^TM^ flow cytometer (High speed FACSAria digital cell sorter, BD Biosciences, Franklin Lakes, NJ, USA). A yellow-green laser of 561 nm was used for the tdTomato detection and a violet laser of 405 nm was used for DAPI detection. Thereafter, enriched cells were seeded for the correct expansion of both cell subpopulations. Importantly, the stem-like phenotype of flow-sorted CSC tdTomato+ cells was confirmed before performing further experiments by the increased expression of stemness markers compared to tdTomato- cells. An example of the stem cell-like gene expression profile of enriched CSC and non-CSC subpopulations from the MDA-MB-231 fluorescent model is shown in Appendix A. Similarly, the in vivo tumorigenicity of tdTomato+ cells derived from the HCC1806 cell line is shown in Appendix A.

### 4.9. Post-Treatment CSC Quantification

The direct effect of the drugs on the CSC population was studied by flow cytometry in ALDH1A-tdTomato CSC models. For each cell line, tdTomato+ and tdTomato- cell subpopulations corresponding to the CSC and non-CSC cells, respectively, were maintained at a 1:1 ratio. Cells were seeded on 6-well plates at a density of 200,000 cells per well. After 24 h, the cells were incubated with the selected treatments (individual and in combination) for 72 h. Thereafter, the medium was removed, and the cultures were refed with complete medium to allow cellular recovery in the absence of drug/s for an additional 48 h. Changes in the amount of tdTomato+ were evaluated using the BD LSRFortessa™ Cytometer (BD Biosciences, Franklin Lakes, NJ, USA) and subsequently analyzed using FCS express 7 Flow cytometry software (De novo, Pasadena, CA, USA) to calculate the increase or decrease in the tdTomato+ population.

### 4.10. Quantitative PCR Analysis

Quantitative RT (real time)-PCR was performed to evaluate the relative expression of CSC stemness markers, such as ALDH1A1, ABCG2, ALOX5, CMKLR1, SOX2, Notch4, POU5F1/Oct-4 and Nanog, after treatment with the selected drugs. Briefly, total RNA was extracted using an RNeasy Mini Kit (Qiagen, Hilden, Germany) and 1 μg retrotranscribed to cDNA (Applied Biosystems, San Francisco, CA, USA). The generated cDNA was quantified using SYBR Green chemistry on a 7500 real-time PCR system (Thermo Fisher Scientific, Madrid, Spain). Two housekeeping (GAPDH and β-Actin) genes served as normalization controls for the analysis of the expression levels of the genes of interest using qbase+ software, version 2.0 (Biogazelle, Gent, Belgium). The cycle threshold values generated were used to calculate the fold change in the expression of the gene of interest. The primer sequences were the same as those previously described [13].

### 4.11. Western Blotting

Cells treated for 24 h with selected drug concentrations were harvested and lysed in M-PER mammalian protein extraction reagent (Thermo Fisher Scientific, Madrid, Spain) supplemented with 1X phosphatase and 1X EDTA-free protease inhibitor cocktails (#539134 and #524625; Calbiochem, Sigma-Aldrich, St. Louis, MO, USA). Lysates were cleared by centrifugation (Refrigerated microcentrifuge 5415R, Eppendorf, Hamburg, Germany) at 13,000 rpm for 20 min at 4 °C. Total protein concentration was quantified using the Pierce™ BCA Protein Assay kit (Thermo Fisher Scientific, Madrid, Spain), following the manufacturer’s instructions. Thirty micrograms of protein were loaded into 10% SDS-PAGE gel. After electrophoresis, isolated proteins were transferred to methanol-activated PVDF membranes (Bio-Rad Laboratories, Hercules, CA, USA), which were probed with different antibodies of interest and chemiluminescence was used to detect the expression of the proteins, β-actin (#8457 Cell Signaling Technology, CST), NF-κB p65 (#8242 CST), phospho-NF-κB p65 (#3033 CST), β-catenin (#610153 BD Biosciences), GSK-3-β (#12456 CST) and phospho-GSK-3-β (#9322 CST).

### 4.12. In Vivo Therapeutic Efficacy

Animal care and in vivo procedures were handled in accordance with the protocols for the Care and Use of Laboratory Animals of the Vall d’Hebron University Hospital Animal Facility. In addition, the Animal Experimentation Ethical Committee at the institution approved the experimental procedures. (ref. CEA-OH/9467/2). All in vivo studies were performed by the ICTS “NANBIOSIS” of CIBER-BBN’s In vivo Experimental Platform for Functional Validation & Preclinical Research area (CIBBIM-Nanomedicine, Barcelona, Spain). Briefly, 1·10^6^ MDA-MB-231.Fluc-GFP cells were suspended in a 1:1 mixture of culture medium and Matrigel (BD Biosciences, Franklin Lakes, NJ, USA) and injected into the mammary fat pad as previously described [57] in six-week-old female NOD/SCID mice (NOD.CB-17-Prkdcscid/Rj), obtained from Janvier Laboratories (France). When tumor volumes reached 75–80 mm^3^, mice were randomized into 3 groups (n = 6–9/each group). Paclitaxel at 10 mg/kg was administered intravenously by tail vein injection 3 times per week in the first week and two times/week the remaining 2 weeks. Niclosamide at 10 mg/kg was given by intraperitoneal injection 5 times per week for 3 weeks. A vehicle of NCS (DMSO: Cremophor^®^ EL, 1:1) was administered intraperitoneally 5 times/week. Tumor growth was monitored twice a week by conventional caliper measurements D × d^2^/2, where D is the major diameter and d is the minor diameter. Upon reaching the endpoint, mice were treated with their corresponding treatment and 1 h after administration were euthanized, and blood was collected by cardiac puncture for the analysis of CTC content by flow cytometry.

### 4.13. Detection of Lung Metastasis

To measure the extent of lung metastasis ex vivo, bioluminescence imaging (BLI) was performed using the IVIS Spectrum (PerkinElmer, Waltham, MA, USA) after administering 150 mg/kg of D-luciferin (Thermo Fisher Scientific, Madrid, Spain) to mice. Afterwards, tumors and tissues (lungs, kidney, spleen and liver) were harvested and weighted. Immediately after necropsy, lung tissues were placed individually into separate wells containing 300 μg/mL of D-luciferin, and imaged and quantified using Living Image^®^ 4.5.2 software (PerkinElmer, Waltham, MA, USA).

### 4.14. Detection of Circulating Tumor Cells from Mice Blood Samples

For isolation of circulating tumor cells (CTCs), blood samples were drawn from each animal by cardiac puncture and further processed using the following protocol [16]. First, the collected samples were subjected to several cycles of erythrocyte lysis using a lysis buffer, which consisted of a mixture of 90% of 0.16 M NH_4_Cl and 10% of 0.17 M Tris (pH 7.65). A total of 5 mL of lysis buffer was added to each tube and samples were incubated for 5 min at 37 °C. Following incubation, the samples were then centrifuged (500 g for 10 min at 4 °C). These steps were repeated until a white cell pellet was obtained, thus indicating the proper lysis of red blood cells. As inoculated tumor cells expressed the GFP fluorescent marker, the resulting pelleted cells were resuspended in a cytometry buffer and examined by flow cytometry. All samples were processed and analyzed using the same parameters. Subsequently, the number of GFP+ cells/mL blood collected was calculated and further normalized by tumor weight from excised tumors at the endpoint.

### 4.15. Statistical Analysis

In vitro experiments were repeated at least 3 times, each involving at least two technical replicates. The in vivo experiment was conducted once with ≥6 animals per group. Data were plotted as the mean value ± SEM (standard error of the mean) and analyzed using GraphPad Prism 6 software. One-way ANOVA analysis, unpaired two-tailed student’s *t*-test or equivalent non-parametric tests were used to investigate the differences between tested compounds and controls. Differences were considered statistically significant when the *p*-value was equal to or below 0.05 (*), 0.01 (**), 0.001 (***) and 0.0001 (****).

## 5. Conclusions

In conclusion, 8Q and NCS anti-CSC drugs displayed synergistic anti-proliferation activity with PTX when combined at specific ratios in TNBC cells. The simultaneous use of PTX with 8Q or NCS combined therapy also increased their anti-CSC potential effect in vitro by inhibiting the inhibition of the NF-κB and Wnt/β-catenin signaling pathways, resulting in the reduction of the mammosphere growth and viability of TNBC cells. Furthermore, the combined therapy of PTX with NCS induced the regression of triple-negative breast tumors and reduced metastasis generation in vivo. These data support the combination of PTX and NCS as a potential treatment for advanced TNBC.

## Figures and Tables

**Figure 1 ijms-23-11760-f001:**
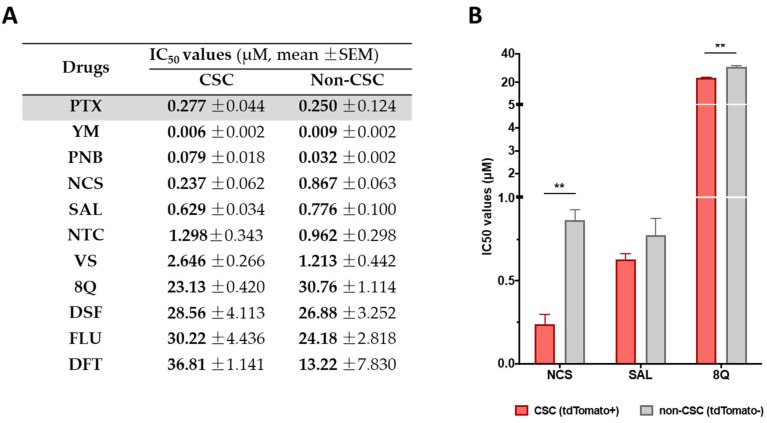
Selective anti-CSC activity of screened drugs in CSC and non-CSC MDA-MB-231 cells grown under attachment conditions. (**A**) IC_50_ values for each compound in both subpopulations after 72 h of incubation. (**B**) Representation of IC_50_ of compounds that showed a greater cytotoxic effect against CSCs compared to non-CSCs. Differences were statistically significant (** *p* < 0.01) when comparing IC_50_ values of 8Q and NCS between both cell subpopulations, but not in the case of SAL. Data are represented as the mean ± SEM of three independent experiments.

**Figure 2 ijms-23-11760-f002:**
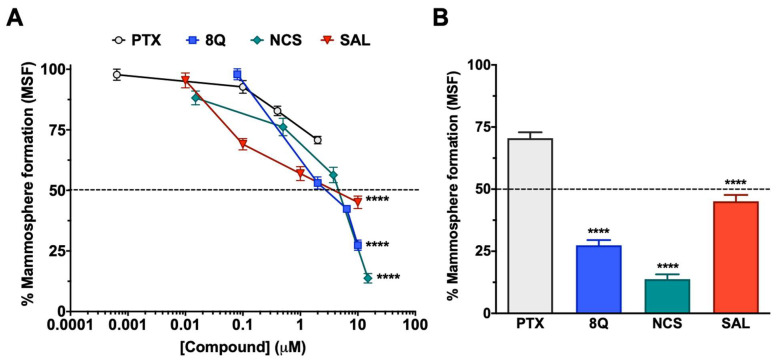
Anti-CSC activity of selected compounds in the MDA-MB-231 CSC population cultured under low attachment conditions. (**A**) Dose-response curves of the CSC mammosphere-forming efficiency (MSF) of MDA-MB-231 cells after drug treatments. (**B**) Percentage of mammosphere after 7-day incubation with 2 µM PTX, 10 µM SAL and 8Q, and 15 µM NCS. Statistically significant results were obtained for all tested compounds when compared to PTX. Data are represented as the mean ± SEM of three independent experiments. **** *p* < 0.0001.

**Figure 3 ijms-23-11760-f003:**
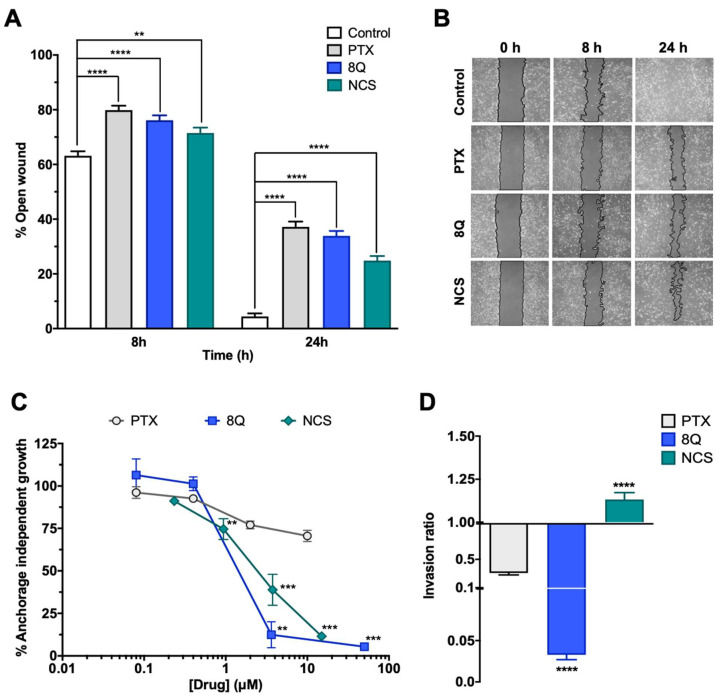
Efficacy of 8Q and NCS inhibiting migration, tumorigenicity and invasion in MDA-MB-231 cell line. (**A**) Wound healing assay showing the percentage of open wounds after 8 and 24 h of treatment with 1 µM PTX, 25 µM 8Q and 0.5 µM NCS. (**B**) Representative images of the wound closing over time. The wound area has been outlined in black and pseudocolored in gray. (**C**) Cell growth in soft agar (anchorage-independent growth) upon treatment with different concentrations of PTX, 8Q and NCS. (**D**) Rate of cell invasion in Matrigel after 24 h treatment of 1 µM PTX, 25 µM 8Q and 0.5 µM NCS. All graphs show the mean ± SEM of at least three independent experiments. All statistical analyses were performed compared to the control non-treated cells (** *p* < 0.01, *** *p* < 0.001, **** *p* < 0.0001).

**Figure 4 ijms-23-11760-f004:**
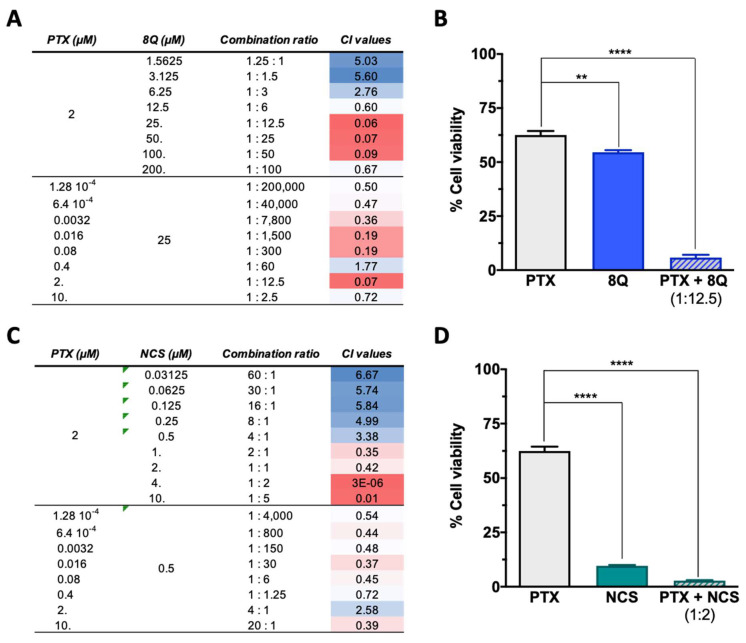
Effect of the combination of 8Q and NCS with PTX on the cell viability of MDA-MB-231 cells. (**A**) Combination index (CI) of different PTX to 8Q drug ratios, shown as heat maps (CI > 1 indicating antagonism in blue, CI < 1 showing synergism in red). Studies were done fixing the PTX concentration first at its IC_50_ value (2 µM) and changing the 8Q concentration (top values), fixing the 8Q concentration at its IC_50_ value (25 µM) and varying the PTX concentration (bottom values). (**B**) Viability of MDA-MB-231 cells treated with PTX, 8Q or the 1:12.5 PTX:8Q ratio. (**C**) CI of different PTX to NCS ratios, again fixing first the PTX at its IC_50_ value (2 µM) and changing the NCS concentration (IC_50_ value 0.5 µM) and vice versa (lower part of the table). (**D**) Viability of MDA-MB-231 cells treated with PTX, NCS or the 1:2 PTX:NCS ratio. Data are represented as the mean ± SEM of three independent experiments (** *p* < 0.01, **** *p* < 0.0001).

**Figure 5 ijms-23-11760-f005:**
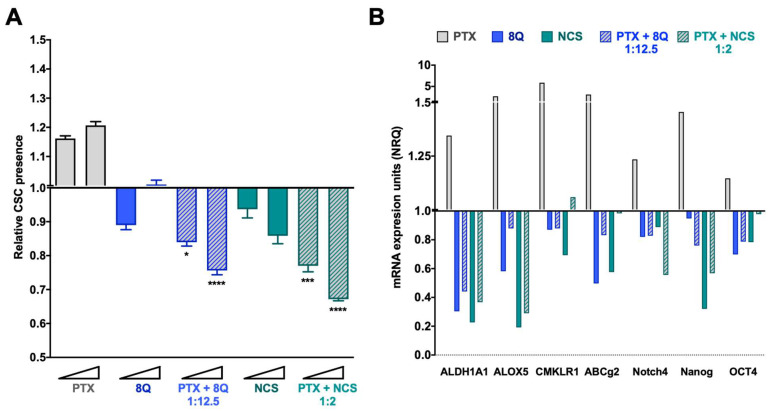
Anti-CSC activity of 8Q and NCS in combination with PTX in the MDA-MB-231 fluorescent CSC model. (**A**) Relative CSC-tdTomato+ presence determined by flow cytometry and referred to control condition. Values below than, equal to or above 1 indicate reduction, maintenance or increase of CSC-tdTomato+ cells within population, respectively. Data are represented as the mean ± SEM of three independent experiments. Statistic *t*-test analyses were performed to compare drug combinations with single anti-CSC drug treatments at the corresponding equivalent drug dose (* *p* < 0.05, *** *p* < 0.001, **** *p* < 0.0001). (**B**) Changes in the stem cell gene expression profile determined by quantitative RT-PCR. Results are expressed as normalized relative quantities (NRQ) and referred to as control conditions. Concentrations tested for drugs were 0.5 and 1 μM for PTX, 6.25 and 12.5 μM for 8Q, 1 and 2 μM for NCS and the corresponding combined ratios.

**Figure 6 ijms-23-11760-f006:**
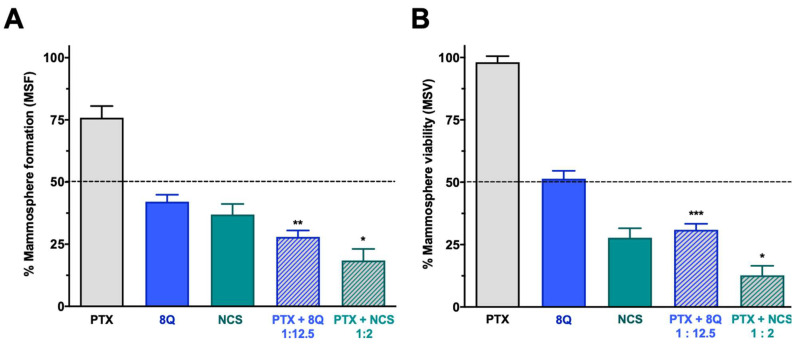
Effect of the combination of 8Q or NCS with PTX in MDA-MB-231 mammospheres. (**A**) Mammosphere-forming (MSF) efficiency of PTX, 8Q, NCS and their combination (2 μM, 25 μM and 4 μM, respectively). (**B**) Mammosphere viability (MSV) of PTX, 8Q, NCS and their combination (4 μM, 50 μM and 8 μM, respectively). Results are represented as the mean ± SEM of three independent experiments and referred to as the non-treated control condition. Statistical *t*-test analysis was performed comparing combination therapy with single anti-CSC treatments at the corresponding equivalent drug dose (* *p* < 0.05, ** *p* < 0.01, *** *p* < 0.001).

**Figure 7 ijms-23-11760-f007:**
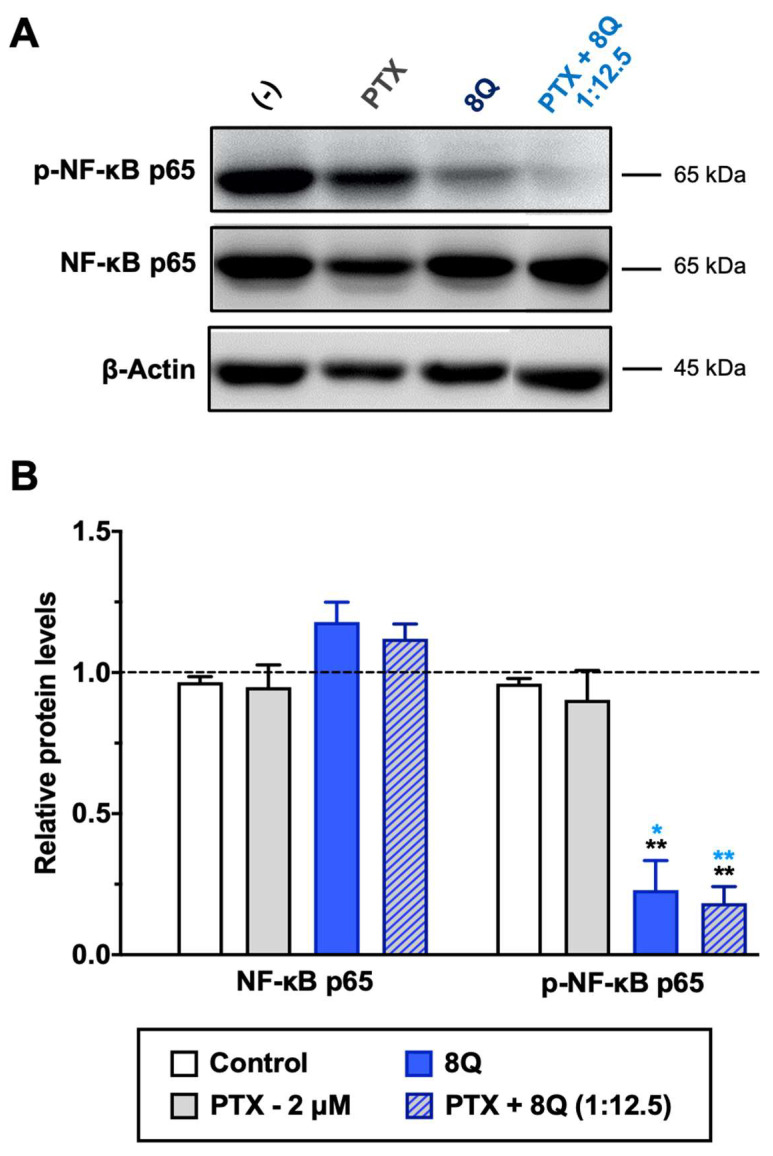
Effect of 8Q and PTX treatments on the NF-κB signaling pathway in MDA-MB-231 cells. (**A**) Representative Western blots of total and phosphorylated NF-κB p65 protein levels upon treatment with PTX, 8Q or their combination (2 μM and 25 μM, respectively). The β-actin protein expression level was used as a loading control. (**B**) Quantification of band intensity in Western blots. The results are expressed as normalized protein levels, referred to β-actin expression, represented as mean ± SEM of three independent experiments. Statistic *t*-test analysis of 8Q and combined therapy was done in comparison with control non-treated cells (black asterisks, ** *p* < 0.01) as well as with PTX treatment (blue asterisks, * *p* < 0.05 and ** *p* < 0.01).

**Figure 8 ijms-23-11760-f008:**
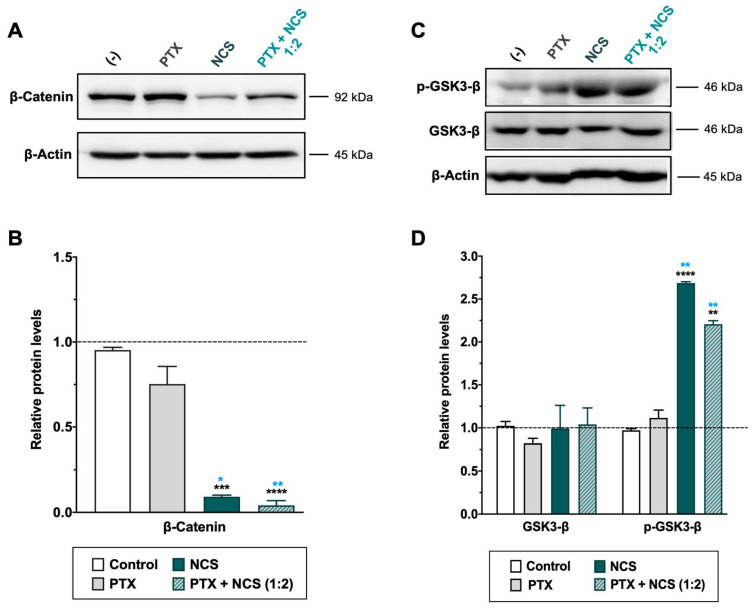
Effect of NCS and PTX treatments in the Wnt/β-Catenin signaling pathway in MDA-MB-231 cells. β-Catenin, total- and phosphorylated-GSK3-β protein levels expression levels after PTX and NCS treatment, both alone and in combination (1:2 ratio), were determined. (**A**–**C**) Representative β-Catenin and GSK3-β protein levels after drug treatments. The β-actin protein expression level was used as a loading control. (**B**–**D**) Graphs represent the quantification band intensity signal referred to β-actin expression, represented as mean ± SEM of three independent experiments. Statistic *t*-test analysis of NCS and combined therapy was performed in comparison with control non-treated cells (black asterisks, ** *p* < 0.01, *** *p* < 0.001 and **** *p* < 0.0001) as well as with PTX treatment (blue asterisks, * *p* < 0.05 and ** *p* < 0.01).

**Figure 9 ijms-23-11760-f009:**
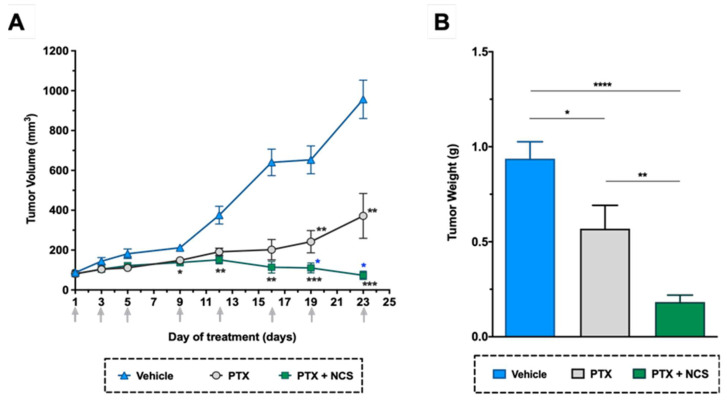
In vivo efficacy of PTX in combination with NCS in mice with orthotopic MDA-MB-231 tumors. (**A**) Tumor volume measurements in the study groups during treatment. (**B**) Tumor weights at the end of the experiment. Arrows indicate treatment points. The results are represented as the mean ± SEM in both graphs (animals/group ≥ 6). Statistic *t*-test analyses of combined treatment were performed in comparison with vehicle (black) as well as with PTX single treatment (blue). In all cases, * *p* < 0.05, ** *p* < 0.01, *** *p* < 0.001 and **** *p* < 0.0001.

**Figure 10 ijms-23-11760-f010:**
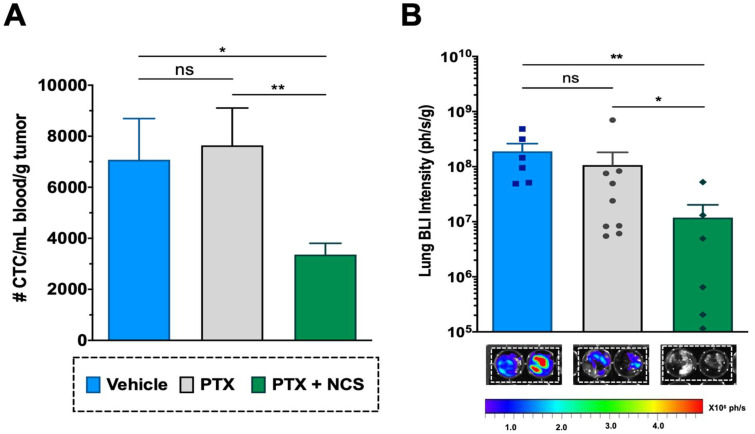
Effect of the PTX-NCS combination on CTC and lung metastasis. (**A**) Quantification of plasma circulating tumor cells (CTCs) isolated from the bloodstream of MDA-MB-231 tumor-bearing mice and further analyzed by flow cytometry. The results are represented as the number of CTC events (mean ± SEM) per mL of blood collected and further normalized per gram of tumor (n ≥ 6). A statistic *t*-test analysis was performed to compare the results between the study groups. (**B**) Quantification of BLI signal intensity from mice lungs of all three study groups. Results are expressed as the mean ± SEM of the BLI signal (ph/s) per gram of lung tissue (n ≥ 6). Statistical Mann–Whitney analysis was performed comparing the results between the study groups. Representative BLI images of excised mice lungs from all three study groups at endpoint necropsy are also shown. Bioluminescent signals were analyzed by ex vivo imaging using the IVIS Spectrum and further quantified using Living Image^®^ 4.5.2 software. Statistical significance is shown as ns *p* > 0.05, * *p* < 0.05, and ** *p* < 0.01.

**Table 1 ijms-23-11760-t001:** Cytotoxic efficacy of selected compounds assessed by MTT assays. IC_50_ values of the assessed compounds in the luminal A cell line MCF-7 and TNBC cell lines (mean ± SEM). PTX (in gray) was used as a reference compound.

Drugs	IC_50_ Values (µM, Mean ± SEM)					
	MCF-7	MDA-MB-231	HCC-1806	MDA-MB-468	BT-549	BT-20
**PTX**	**0.012** ± 0.003	**1.473** ± 0.014	**0.008** ± 0.002	**0.004** ± 0.001	**0.015** ± 0.007	**0.012** ± 0.004
**YM**	**0.019** ± 0.004	**0.004** ± 0.001	**0.018** ± 0.002	**0.002** ± 0.001	**0.025** ± 0.001	**0.001** ± 0.001
**PNB**	**0.058** ± 0.022	**0.084** ± 0.014	**0.016** ± 0.004	**0.002** ± 0.001	**0.144** ± 0.007	**0.084** ± 0.032
**VS**	**0.193** ± 0.078	**9.365** ± 4.329	---	---	---	---
**NCS ***	**0.373** ± 0.097	**0.545** ± 0.024	**3.106** ± 0.101	**1.878** ± 0.458	**0.960** ± 0.034	**1.547** ± 0.062
**FLU**	**0.787** ± 0.110	**17.87** ± 1.300	**0.402** ± 0.130	**0.001** ± 0.001	**0.504** ± 0.049	**0.041** ± 0.003
**NTC**	**1.055** ± 0.166	**1.314** ± 0.159	---	---	---	---
**DSF**	**1.182** ± 0.152	**11.65** ± 0.253	---	---	---	---
**SAL ***	**2.702** ± 0.107	**0.399** ± 0.149	---	---	---	---
**DFT**	**3.242** ± 1.180	**32.10** ± 1.720	---	---	---	---
**8Q ***	**4.442** ± 0.137	**25.55** ± 0.151	**25.19** ± 1.08	**3.295** ± 0.111	**79.74** ± 2.890	**16.15** ± 0.67
**EVE**	**10.38** ± 0.591	**13.21** ± 1.449	---	---	---	---
**GLA**	**21.71** ± 4.367	**18.72** ± 5.362	---	---	---	---
**ISO**	**25.11** ± 0.724	**53.33** ± 5.275	---	---	---	---
**6-SHO**	**27.73** ± 1.597	**24.08** ± 2.995	---	---	---	---
**MET**	**3523** ± 601	**3315** ± 396	---	---	---	---
**ACE**	**4670** ± 417	**6806** ± 842	---	---	---	---
**CIT**	**>200**	**>200**	---	---	---	---

Drug abbreviations stand for: PTX, paclitaxel; YM, YM-155 hydrocloride; PNB, panobinostat; VS, VS-5584; NCS, niclosamide; FLU, flubendazole; NTC, nitidine chloride; DSF, disulfiram; SAL, salimomycin; DFT, defactinib; 8Q, 8-quinolinol; EVE, everolimus; GLA, glabridin; ISO, isoliquiritigenin; 6-SHO, 6-shogaol; MET, metformin hydrochloride; ACE, acetaminophen; CIT, citral. Compounds proceeding into further assays are labeled with an asterisk *.

## Data Availability

The datasets used and/or analyzed during the current study are available from the corresponding author on reasonable request. Appendix A consist of 1 table and 8 figures and are fully available for readers.

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
