# Peer review of "Selectively Targeting Breast Cancer Stem Cells by 8-Quinolinol and Niclosamide"

_ijms, 2022, doi:10.3390/ijms231911760_

Round 1
Reviewer 1 Report
In this manuscript by Camara-Sanchez et al, the researchers provide more evidence that 8-quinolinol and niclosamide (8Q and NCS, respectively) can independently inhibit the growth of cancer stem cells derived from multiple breast cancer cells lines. These results are shown in cell viability assays, mammosphere formation tests, healing assays and in vitro invasion assays. Importantly the inhibitory effects are more pronounced compared to paclitaxel (PTX), and combined therapies (either PTX plus NCS or 8Q) exhibit synergistic inhibition. They confirm inhibition signaling pathways known to modulated by 8Q and NCS, NF-kB and Wnt/b-catenin, respectively. Novelty of their results include suppression of several genes commonly identified in CSCs and associated with stem cells. Perhaps the most intriguing result is the reversal of CSC formation caused by PTX, which presents a significant challenge when treating breast cancer with this chemotherapeutic. The authors then present results from in vivo studies of mice bearing NDA-MB-231-derived tumors that show decreased tumor volume and weight with PTX plus NCS treatment, and reduced circulating tumor cells in the mice blood. The results support their conclusion that NCS and perhaps 8Q should continue to be studied for their potential to treat breast cancer, in particular, triple negative breast cancer. There are some issues that require attention, and some grammatical errors that should be easy to correct.
Page 6 (text) and 7 (Table 1), the authors should indicate units in the table legend for all IC50 values (assume microM concentrations). In addition, the drug with high cytotoxicity potential could be emphasized with a marker to guide the reader to those drugs discussed in the text.
Figure 1A, again the unit of concentration should be indicated in the table or legend.
Page 8 (and applicable to figures thereafter), the authors should indicate why they chose to only test up to 1 microM PTX, vs. the higher concentrations used for 8Q, NCS and SAL. Of note, they used 10 microM in the next figure (Fig 3).
Page 12, second paragraph, the authors could add some values to the descriptions of the supplemental Figures S5A and C. In particular values should be added for anti-CSC effect of 8Q an NSC when administered with PTX, in order to support the term "significantly enhanced", such as fold-increases in growth inhibition, and specific results as to how the two alternative cell lines compared to MDA-MB-231 in the last sentence.
Page 16, when introducing the in vivo studies with NCS + PTX vs. PTX alone, the authors mention that they focused on NCS based on its higher anti-CSC activity, but the in vitro data still indicate effective growth inhibition by 8Q, so why did they not test this with PTX in their in vivo studies? Perhaps a better justification for just focusing on NCS would suffice.
In addition to the transcriptional regulators analyzed, did the authors check for any effects of either drug independently, or in combination with PTX, on apoptosis regulators/signaling pathways? NCS has been shown to affect caspase-3 and BCL-2 activities, among other apoptosis regulators, so this would be excellent to show as a follow-up to the growth inhibition studies, in order to understand if there is also evidence of apoptosis in the treated CSCs.
Page 19, first paragraph, second sentence, the authors indicate that they showed both 8Q and NCS inhibited CSC hallmarks, and indeed showed loss of migration and invasion (albeit with an in vitro assay), but they did now show that the drugs inhibited "neoplastic transformation", which would require showing that the drugs actually transformed oncogenic potential of cancer cells either in vitro or in vivo. They do show changes in gene expression profiles, but these are focused on "stemness" genes, not oncogenes. This sentence therefore needs to be edited.
Page 20, under Conclusions, second sentence, the authors did not actually show that the anti-CSC effects of teh the drugs was due to inhibition of NF-kB or Wnt/B-catenin signaling pathways, therefore this sentence should be edited to indicate that anti-CSC potential may be due to inhibition of these pathways (a direct effect would require further analyses to show a causitive effect).
Grammatical errors (this is not comprehensive):
Page 7, sole paragraph, second sentence, there is a grammatical error - suggest adding after "expression of stemness marker" a linker to "further subjected to conventional proliferation", such as "and then were....."
Page 8 paragraph, 4th sentence, the grammatical error "but significantly higher than the produced...." can be corrected by simply changing "the" to "that" (i.e., "than that produced by the...").
Page 12, third paragraph, third sentence beginning with "While 8Q and NCS treatments....." the sentence is grammatically incorrect.
Page 20, top paragraph, second sentence, the phrase "8Q and NCS acted inhibiting....." needs rewording, perhaps change to "acted to inhibit....".
Author Response
"Please see the attachment."

Reviewer 2 Report
This paper aimed to identify potential drugs targeting CSCs and to explore the efficiency of their combination with standard chemotherapy for TNBC. Treatment. Their study revealed that the potential drug 8Q and NCS showed remarkable specific anti-CSC activity in vitro. And, they also show that the combination of NCS with PTX reduced tumor growth, and reduced the number of circulating tumor cells and the incidence of lung metastasis.
Comments:
1. Breast CSCs were usually obtained from sorted CD44+CD24- or ALDH+ subpopulation or enriched mammospheres either from cultured cell lines or fresh surgical tissue. In this paper, the author adopted a ALDH1A1:tdTomato CSC models, in which CSC can be identified by the expression of tdTomato fluorochrome. However, for CSCs identification and functional assay, excepted for mammospheres and stemness gene expression, more functional CSC assays including CSCs ratio, chemo-resistance and more importantly, in vivo tumor formation are still needed.
2. In their mammosphere formation assay in ultra-low attachment 96-wells plates, higher cell density of 1,000 viable cells/well were inoculated. In such a high density, how did the author exclude the aggregates? The author should provide the images of mammospheres.
3. In Figure10A, the quantification of plasma circulating tumor cells (CTCs) isolated from the bloodstream of MDA-MB-231 tumor-bearing mice was shown. Is the title of longitudinal coordinates showed the relative value for CTC? What did the number of CTC normalized to per gram of tumor tissue excised stand for? Please clarify.
4. The author found that SAL, a generally considered selective anti-CSCs drugs, did not show differences of IC50 values between CSCs and non-CSCs. And, the effect on MSF was remarkably lower. Could the author explain this in the discussion?
Finally, thanks for the opportunity to review this manuscript.
Author Response
"Please see the attachment."

Reviewer 3 Report
1. The authors in the manuscript have worked on potential drugs targeting CSCs to be further employed in combination with standard chemotherapy in TNBC treatment. Among all tested drug candidates, 8Q and NCS showed remarkable specific anti-CSC activity in terms of CSC viability, migration, invasion and anchorage independent growth reduction in vitro. The authors also revealed the solely use of PTX that increased the relative presence of CSCs in TNBC cells but the combination with 8Q and NCS counteracted this pro-CSC activity of PTX whilst significantly reducing cell viability. The manuscript has been conceptualized well; however, a few modifications may help improve the quality of the manuscript.
2. Notch, Wnt/β-catenin, PI3K/Akt/mTOR, NF-κB, HIF and STAT3 signaling pathways should be explained.
3. The author used 17 drugs for the study. What is the criteria for selecting the drugs and Why only 17 drugs?
4. The author can check the paper thoroughly for any typographical errors.
5. In methods, the paragraph on pharmacological agents is poorly referenced. Also provides references in methods.
6. In page 8, the author has mentioned Figure 1 part A as figure, kindly changed it into Table.
Author Response
"Please see the attachment."

Round 2
Reviewer 1 Report
The authors have addressed the concerns of this reviewer and improved the impact of the manuscript. There are some minor typographical errors in the manuscript, so a careful review is warranted. For example, the last sentence of the abstract has "fpr" instead of "for".